# Bionomic responses of *Spodoptera frugiperda* (J. E. Smith) to lethal and sublethal concentrations of selected insecticides

**Kokou Rodrigue Fiaboe**[1,2¤]*, **Ken Okwae Fening**[1,2], **Winfred Seth Kofi Gbewonyo**[1,3], **Sharanabasappa Deshmukh**[4]

**1** African Regional Postgraduate Programme in Insect Science (ARPPIS), School of Agriculture, College of Basic and Applied Sciences, University of Ghana, Legon, Accra, Ghana, **2** Soil and Irrigation Research Centre (SIREC), School of Agriculture, College of Basic and Applied Sciences, University of Ghana, Legon, Accra, Ghana, **3** Department of Biochemistry, Cell and Molecular Biology, School of Biological Science, College of Basic and Applied Science, University of Ghana, Legon, Accra, Ghana, **4** Department of Entomology, College of Agriculture, Keladi Shivappa Nayak University of Agricultural and Horticultural Sciences (UAHS), Shivamogga, Karnataka, India

¤ Current address: Department of Zoology and Entomology, University of Pretoria, Hatfield, South Africa
* rfiaboe@yahoo.com

**Data Availability Statement:** All relevant data are within the manuscript and its Supporting Information files.

**Funding:** The authors received no specific funding for this work.

## Abstract

Since 2016, the invasive insect *Spodoptera frugiperda* (J. E. Smith) (Lepidoptera: Noctuidae) from the Americas has made maize production unattainable without pesticides in parts of Sub-Saharan Africa and Asia. To counteract this pest, farmers often resort to the use hazardous pesticides. This study aimed to investigate botanicals, microbials, and semi-synthetic insecticides in Ghana for pest control without harming local ecosystems. Under laboratory and on-station conditions, the present study evaluated the acute and sublethal responses of *S. frugiperda* to: (i) *Pieris rapae* Granulovirus (*Pr*GV) + *Bacillus thuringiensis* sub sp. *kurstaki* (*Btk*) 5 WP, (ii) *Btk* + monosultap 55 WP, (iii) ethyl palmitate 5 SC, (iv) azadirachtin 0.3 SC, (v) acetamiprid (20 g/l) + λ-cyhalothrin (15 g/l) 35 EC, (vi) acetamiprid (30 g/l) + indoxacarb (16 g/l) 46 EC, and (vii) emamectin benzoate 1.9 EC. The results showed that at 96 hours post-exposure emamectin benzoate-based formulation has the highest acute larvicidal effect with lower $LC_{50}$ values of 0.019 mL/L. However, the results suggested strong sublethal effects of *Pr*GV + *Btk*, azadirachtin, and ethyl palmitate on the bionomics of *S. frugiperda*. Two seasons on-station experiments, showed that the semi-synthetic emamectin benzoate and the bioinsecticide *Pr*GV + *Btk* are good candidates for managing *S. frugiperda*. The promising efficacy of emamectin benzoate and *Pr*GV + *Btk* on the bionomics of *S. frugiperda* in the laboratory and on-station demonstrated that they are viable options for managing this pest.

## Introduction

Agricultural productivity plays a crucial role in determining the economic growth of a country. In sub-Saharan Africa, agriculture is a significant contributor to the GDP of countries and

**Competing interests:** The authors have declared that no competing interests exist.

employs a large proportion of the population [1,2]. For instance, in Ghana, agriculture accounts for 20% of the GDP of the country and is vital for food security, with maize (*Zea mays* L., Poales: Poaceae) being a major crop [2,3]. Maize is a staple crop in Ghana that can be cultivated in almost all ecological zones of the country [2,3]. However, agricultural production in Ghana is under threat from various pests, including the fall armyworm (*Spodoptera frugiperda*, Lepidoptera: Noctuidae) [4,5]. *Spodoptera frugiperda* is an invasive pest, native to the Americas, that was first detected in Africa in early 2016 and has since spread rapidly across the continent, posing a serious threat to food security and agricultural livelihoods [6–8]. Infesting other staple crops like sorghum, millet, rice, and wheat, *S. frugiperda* is a voracious feeder that predominantly targets maize crops [9–11]. Clusters of eggs are laid by female moths, and the resulting larvae are extremely destructive [6,12]. The larvae of the pest feed on the foliage of host plants and possess the ability to tunnel into maize cobs, resulting in significant reductions in both grain quantity and quality [6,13]. The fast reproduction rate, preference for maize, and pesticide tolerance of *S. frugiperda* make it a challenging pest to manage [14,15].

Farmers have turned to synthetic insecticides for controlling the pest, which are hazardous to the environment and human health [4,16]. Studies have shown that the use of synthetic insecticides can lead to resistance development, environmental pollution, and health hazards [15,17]. As a result, there is a growing interest in alternative pest control methods such as botanicals, microbials, and insect growth regulator pesticides, which are considered safer for the environment and human health, and less likely to induce resistance [5,18–20].

In response to the urgent *S. frugiperda* infestation in Ghana, farmers were provided with insecticides, encompassing synthetic, botanical, and microbial formulations [21–23]. However, it is important to highlight that these distributions took place without conducting prior efficacy testing or determining suitable application doses. Instead, the application doses were approximated based on those used for similar Lepidopteran pests affecting other crops [4,5]. However, it is important to note that variations in environmental conditions, biological factors, and insecticide resistance can affect the sensitivity of pesticides [24,25]. Furthermore, to ensure the safety of non-target species and the effectiveness of the pesticides, bioassays should be conducted before approving new chemical formulations for use by farmers, as recommended by WHO [26]. Meanwhile, traditional approaches of bio-efficacy testing can be challenging due to their narrow focus on individual mortality in a short term and the variation in the mode of action of active ingredients [27]. For instance, an insecticide may not induce acute mortality, but its sublethal toxicity can still significantly impact insect populations, including the development of resistance, disruption of behavior, and reduced reproductive success [27]. Thus, understanding the sublethal effects of insecticides is crucial for developing effective pest management strategies that minimize chronic impacts on non-target organisms and reduce the development of resistance in pest populations.

On-station and/or field trials are vital to ensuring that the recommended doses of insecticides are the minimum necessary to control the target pests effectively, while minimizing non-targeted effects, such as phytotoxicity [28–30]. Additionally, on-station conditions typically represent more realistic conditions compared to laboratory studies, providing more reliable data for making management decisions [28,30]. Insects in the field are exposed to a range of environmental factors that can affect their behavior and susceptibility to insecticides [30]. In the present study, we hypothesized that not all the insecticide formulation in use or recommended to maize farmers for the management of *S. frugiperda* may demonstrate promising acute efficacy against the pest. Therefore, the primary objective of this study was to assess the efficacy of selected commercial pesticides against *S. frugiperda* in Ghana. The study was conducted in two phases, where lethal concentrations of selected commercial pesticides were determined under laboratory conditions, and the effects of sublethal concentration on the

bionomics, reproduction parameters, and longevity of *S. frugiperda* were investigated. In the second phase, the effectiveness of selected commercial pesticides in reducing the infestation of *S. frugiperda* and increasing maize grain yield was evaluated under field conditions.

Understanding the sublethal effects of insecticides is crucial for developing effective pest management strategies that minimize the impact on non-target organisms and reduce the development of resistance in pest populations. The findings of this study will provide valuable insights into the efficacy of selected commercial pesticides and their impact on *S. frugiperda*, contributing to the development of more effective and sustainable pest management strategies for farmers in Ghana.

## Materials and methods

### Ethics statement

This study was conducted outside of national parks or any protected areas. The crop used in the study, maize (*Zea mays* L.), and the invasive insect pest, the fall armyworm (*Spodoptera frugiperda*, Lepidoptera: Noctuidae), are not considered endangered or protected species.

### Laboratory studies

**Experimental insect and plant.** We obtained test-insects from a colony of *S. frugiperda* at the Laboratory of Entomology of SIREC. The colony was established in 2017 by collecting egg masses and larvae from insecticide-free maize fields across Ghana during a national inventory survey of the natural enemies of the pest. To rear the insects, we followed the method described by [12] and fed the larvae fresh castor bean leaves and adults a 10% honey solution. We tested the insects under conditions of 27 ± 1˚C temperature, 60 ± 5% RH, and 12 hours of photophase. To conduct the larvicidal bioassays, we planted "QPM var. Obatanpa" maize in plastic pots (8 cm diameter × 7.5 cm high) containing 0.5 L planting substrate (manure:soil 1:5).

**Insecticides.** Seven commercial insecticides obtained from different sources were tested *in vitro* on the larvae of *S. frugiperda* (Table 1). The insecticide formulations included: Synthetics, Strike 1.9 EC™ (19.2 g/L of emamectin benzoate), K-Optimal 35 EC® (20g/L acetamiprid + 15g/L lambda-cyhalothrin), Viper 46 EC® (16g/L acetamiprid + 30g/L indoxacarb); Botanicals, Adepa 5 SC® (5% ethyl palmitate), Neemgold 0.3 SC® (3% azadirachtin); and microbials Agoo 55 WP® (55%*Bacillus thuringiensis kurstaki* + 45% monosultap), and Bypel 5 WP® [10000 PIB/mg *Peris rapae* Granulosis Virus + 16000 IU/mg *Bacillus thuringiensis kurstaki*).

**Table 1. Details on insecticides assessed and the concentration ranges used.**

| Trade name | Common name[a] | Manufacture | Conc. Recommended | No. of tested conc.[b] | Conc. Ranges[c] |
|---|---|---|---|---|---|
| **Strike 1.9 EC™** | 19.2 g/L Emamectin benzoate | B. Kaakyire Agrochemicals | 1 mL/L | 5 | 0.001–10 |
| **Viper 46 EC®** | 16 g/L Acetamiprid + 30 g/L Indoxacarb | Arysta Life Science Ltd. | 3 mL/L | 5 | 0.03–24 |
| **K-Optimal 35 EC®** | 20 g/L Acetamiprid + 15 g/L ʎ-cyhalothrin | Macrofertil Gh. Ltd. | 3.3 mL/L | 6 | 1.67–40 |
| **Adepa 5 SC®** | 5% Ethyl palmitate | Kwadutsa and Joam Co. Ltd. | 6.7 mL/L | 7 | 3.33–1000 |
| **NeemGold 0.3 SC®** | 3% Azadirachtin | Foliage Crop Solutions Ltd. | 2 mL/L | 6 | 0.50–54 |
| **Bypel 1 WP®** | 10,000 PIB/mg *Pr*GV + 16,000 IU/mg *Btk* | Wuhan UNIOASIS BioTech Co. Ltd. | 1000 mg/L | 6 | 125–4000 |
| **Agoo 55WP®** | 55% *Btk* + 45% Monosultap | Kwadutsa and Joam Co. Ltd. | 3333 mg/L | 5 | 863–13333 |

[a] *Pieris rapae* Granulosis Virus (*Pr*GV); PIB: Polyhedra Infective Bodies; *Btk*: *Bacillus thuringiensis* subsp. *kurstaki*; IU: International Units.

[b] Number of concentrations tested (expressed as either w/v or v/v of product per liter of water).

[c] Range of concentrations tested for each formulated insecticide product.

Conc.: Concentration of product. No.: Number.

Concentrations (w/v or v/v) of the product per liter of water were prepared using distilled water. The baseline concentrations of each insecticide were determined following a preliminary test by exposing batches of 25 third instar larvae per a wide range of concentrations in leaf-dip assay and monitored for 72 h. The minimum lethal concentration ($LC_5$), the maximum lethal concentration ($LC_{95}$), and the median lethal concentrations ($LC_{30}$ and $LC_{70}$) were chosen to establish the number and range of concentrations for each insecticide [26] (Table 1), with the manufacturer-recommended concentrations included in the tests. The concentrations were prepared using serial dilution method [31].

**Larvicidal bioassays.** The lethal concentrations of each formulation were determined using a leaf-dip method with fresh leaves of potted maize plants aged 14 to 18 days. The foliage of maize plant was briefly dipped into each test dilution, along with a control of distilled water, and air-dried for 40 minutes at room temperature. The poisoned maize leaves were then cut into pieces (2.5 cm) and placed in plastic cups (7.3 × 7.3 × 6.0 cm). Batches of 10 early third instar larvae of *S. frugiperda* were individually confined with insecticide-treated leaves, and their death time was monitored up to 96 hours. This constituted the biological replicates and replicated with 8–10 batches of larvae per concentration. Experiments were conducted at 27 ± 1°C, 60 ± 5% RH, and 12:12 h (L:D) photoperiod. When control mortality exceeded 10%, data were corrected using the following Schneider-Orelli's formula [32]:

$$Corrected\ mortality\ (p) = \frac{\%Responded - \%Responded\ in\ Control}{100 - Responded\%\ in\ Control}$$

**Sublethal effects on the bionomics of $F_0$ parent generation of *S. frugiperda*.** To evaluate the effects of sublethal insecticide concentrations on *S. frugiperda* progeny reproduction, maize plants were impregnated with $LC_{25}$ concentrations in distilled water using the same method as the larvicidal bioassays. A control group was treated with distilled water only. Chopped, treated leaves were placed in plastic cups, and third-instar larvae of *S. frugiperda* were individually placed in each cup to feed on either untreated or insecticide-treated maize leaves. After 48 hours of feeding, the insecticide-treated leaves were replaced daily with untreated leaves for the surviving larvae until they emerged as adults. The $F_0$ generation moths were then coupled in a 1:1 sex ratio, with five couples being placed in an oviposition cage (40 × 40 × 55 cm) along with individually potted maize plants. The longevity of the pupae and adults (days), as well as fecundity (i.e., number of eggs produced by individual female insect), were meticulously recorded. The experiment was conducted under controlled environmental conditions of 27 ± 1°C, 60 ± 5% RH, and a 12:12 h (L:D) photoperiod.

**Sublethal effects on the bionomics of $F_1$ generation of *S. frugiperda*.** Ten individual eggs laid by insecticide-survived parents ($F_0$ generation) from the same treatment plant were gently placed on untreated potted maize plant leaves using a tiny brush pen. Each insecticide treatment was replicated 7–9 times, resulting in 7–9×10 eggs per treatment. Upon hatching, the larvae were fed with untreated maize leaves in separated cups until adult emergence. Ten adult moths (1:1 sex ratio) from the same treatment were independently paired on maize plants (in 20 × 20 × 35 cm Plexiglass cages), and the daily oviposition was monitored and recorded by changing the maize plants until the insects died. The duration of each developmental stage was recorded.

## On-station experiments

This research was conducted at the University of Ghana's Soil and Irrigation Research Centre, (SIREC) at Kpong (00 04' E, 60 09' N), in the Lower Manya District of the Eastern Region of

Ghana. SIREC, is located approximately 22 m above sea level and lies within the lower Volta basin of the Coastal Savanna agro-ecological zone of Ghana. It is characterized by an annual rainfall of 700–1100 mm and an average annual temperature of 28˚C. The relative humidity (RH) ranged between 59%-93% throughout the year. The main soil type was the Vertisols (black clay soil). Two consecutive on-station experiments were carried out to assess the effectiveness of four insecticides, namely Strike 1.9 EC™ (emamectin benzoate), Bypel 1 WP® (*Pr*GV + *Btk*), Agoo 55WP® (*Btk* + monosultap), and NeemGold 0.3 SC® (azadirachtin). The selection of these insecticides was based on their larvicidal potency (see Table 1) and also to represent each of the insecticide classes, including synthetic, microbial, and botanical. The first trial took place from September 3, 2018 to December 26, 2018 during the minor rainy season; the second trial was conducted under surface irrigation and was conducted from January 14, 2019 to April 27, 2019 during the dry season.

**Plant material.**   Maize variety "QPM Obatanpa var." was used for the field trials and arranged in a randomized complete block design (RCBD) with four replicates. Each block had 5-unit plots (10 m × 8 m) representing the single factor, insecticide treatments. Two seeds were sown per hill with an interplant distance of 80 cm between rows and 40 cm between plants. The alley between neighboring plots were 2.5 m to avoid spray drift. A complex fertilizer ($N_{15}P_{15}K_{15}$) was applied at 250 kg/ha and supplied with 100 kg N/ha in the form of ammonium nitrate. Standard cropping and agronomic practices were carried out, including chemical control of weeds using selective herbicide 2,4-Dichlorophenoxyacetic acid (2,4 D). Handpicking of resistant weed and alternative host of the pest, mainly crabgrass (*Digitaria* sp.) was performed during both seasons. During the dry season, surface irrigation method was employed to provide 7-day intervals water to the plants.

**Insecticide preparation, application and data collection.**   In both trials, recommended doses were used: Strike 1.9 EC™ (emamectin benzoate) at 1 mL per 1 L of water (150 mL / ha), Bypel 1 WP® (*Pr*GV + *Btk*) at 1 g per 1 L of water (200g / ha), Agoo 55WP® (*Btk* + monosultap) at 3.3 g per L of water (400g / ha), and NeemGold 0.3 SC® (azadirachtin) at 2 mL per 1 L of water (400 mL / ha). Control plots received no insecticide. A 15 L knapsack sprayer with a full-cone nozzle was used for application, calibrated before use to ensure proper flow rate and insecticide dose. Applications were made late afternoon (4–6 pm) beginning 10 days after germination to reduce photodegradation and minimize side effects on pollinators [33]. The number of insecticide applications was decided based on the infestation level determined by visual scouting [12], resulting in two and three treatments during the first and second trials, respectively.

Data collected included alive larvae per plant (larval incidence) and maize grain yield. Ten plants per plot were sampled, and larval incidence data were collected 1 day before and 3, and 7 days after treatment. Yield per hectare was estimated by multiplying the yield per 10 plants per plot by theoretical plant density per hectare.

## Statistical analyses

The laboratory data were analyzed using Finney's Probit Analysis Programme [34] to determine lethal concentrations, fiducial limits (95%), slopes, and Chi-square values. The population and reproduction parameters were calculated using TWOSEX-MSChart 2023 software [35,36], with the standard errors determined by bootstrapping with 100,000 repetitions. Paired bootstrapping was used to evaluate differences between groups ($P < 0.05$). The data were checked for normality using Shapiro–Wilk test and equal variances using Levene test before selecting the appropriate statistical analyses. Thus, the analysis of larval mortality induced by the concentrations recommended by the manufacturers was conducted using the Likelihood

ratio test (LR test) applied to a generalized linear model (GLM). Similarly, pupal duration, mortality (between pupa and adult stages), adult longevity, and fecundity were analyzed using the LR test with GLM. The percent population reduction resulting from insecticide application was determined using the following formula:

$$R = \frac{(\text{PreTP} - \text{PostTP})}{\text{PreTP}} \times 100$$

Here, R represents the percent population reduction, while PreTP and PostTP represent the population densities before and after insecticide spray per treatment plot, respectively. The calculated R values were subjected to statistical analysis using a one-way analysis of variance (ANOVA). In addition, to analyze the differences between the control treatment (no insecticide) and the other treatments (insecticide receiver plots), the Dunnett (two-sided) test was employed [37]. This analysis was conducted with a confidence interval of 95%. Similarly, yield data were analyzed using one-way ANOVA. The means were separated using Tukey's Range Test ($\alpha$ = 0.05). Except for the population and reproduction characteristics, all data were analyzed in R [38].

## Results

### Larvicidal potency of insecticides on *S. frugiperda* larvae mortality in laboratory bioassay

Table 2 (toxicity bioassay results) showed $LC_{50}$ values at 96 HAT (hours after treatment) ranging from 0.019 mL/L Strike 1.9 EC™ (emamectin benzoate) to 108.5 mL/L Adepa 5 SC® (ethyl palmitate) with slope values below 1 observed for emamectin benzoate and ethyl palmitate. According to the findings presented in Tables 1 and 2, the $LC_{50}$ values for emamectin benzoate, Viper 46 EC® (acetamiprid + indoxacarb), Bypel 1 WP® (*Pr*GV + *Btk*), and Agoo 55WP® (*Btk* + monosultap) were lower than the dosages recommended by the manufacturer. In contrast, the $LC_{50}$ of K-Optimal 35 EC® (acetamiprid + γ-cyhalothrin), Adepa 5 SC® (ethyl palmitate), and NeemGold 0.3 SC® (azadirachtin) were higher than the recommended dose (Tables 1 and 2). However, it is noteworthy that the calculated $LC_{90}$ values for all of the insecticides tested were higher than the manufacturer-recommended dosages (Tables 1 and 2).

**Table 2. Dose-mortality responses of *S. frugiperda* larvae to different insecticides at 96 HAT.**

| Insecticides | N | $LC_{10}$ (95% F.L.) | $LC_{50}$ (95% F.L.) | $LC_{90}$ (95% F.L.) | Slope ± SE | df | $\chi^2$ |
|---|---|---|---|---|---|---|---|
| **Emamectin benzoate** | 575 | 0.0000587 (1.03E-05–3.34E-04) | 0.019 (0.003–0.108) | 6.09 (1.07–34.69) | 0.513±0.385 | 4 | 0.8 |
| **Acetamiprid + Indoxacarb** | 585 | 0.17 (0.08–0.35) | 1.78 (0.86–3.65) | 18.445 (8.99–37.82) | 1.274±0.159 | 4 | 0.49 |
| **Acetamiprid + ʎ-Cyhalothrin** | 665 | 1.41 (0.89–2.23) | 7.39 (4.66–11.71) | 38.69 (24.40–61.34) | 1.786±0.102 | 5 | 0.9 |
| **Ethyl palmitate** | 740 | 2.83 (1.13–7.10) | 108.51 (43.28–272.03) | 4152.48 (1656.48–10410.36) | 0.819±0.204 | 6 | 0.42 |
| **Azadirachtin** | 700 | 0.48 (0.25–0.93) | 4.73 (2.48–9.04) | 45.92 (24.05–87.68) | 1.304±0.143 | 5 | 0.89 |
| **_Pr_GV + _Btk_** | 650 | 51.06 (31.20–83.57) | 257.46 (157.32–421.34) | 1298.81 (793.20–2124.33) | 1.828±0.109 | 5 | 0.97 |
| **_Btk_ + Monosultap** | 650 | 706.94 (466.93–1070.33) | 2766.16 (1827.02–1827.02) | 10823.52 (7148.82–16387.13) | 2.255±0.092 | 4 | 0.32 |

N: Number of larvae tested. F.L.: Fiducial Limits. LC: Lethal concentration 10% ($LC_{10}$), lethal concentration 50% ($LC_{50}$), lethal concentration 90% ($LC_{90}$).

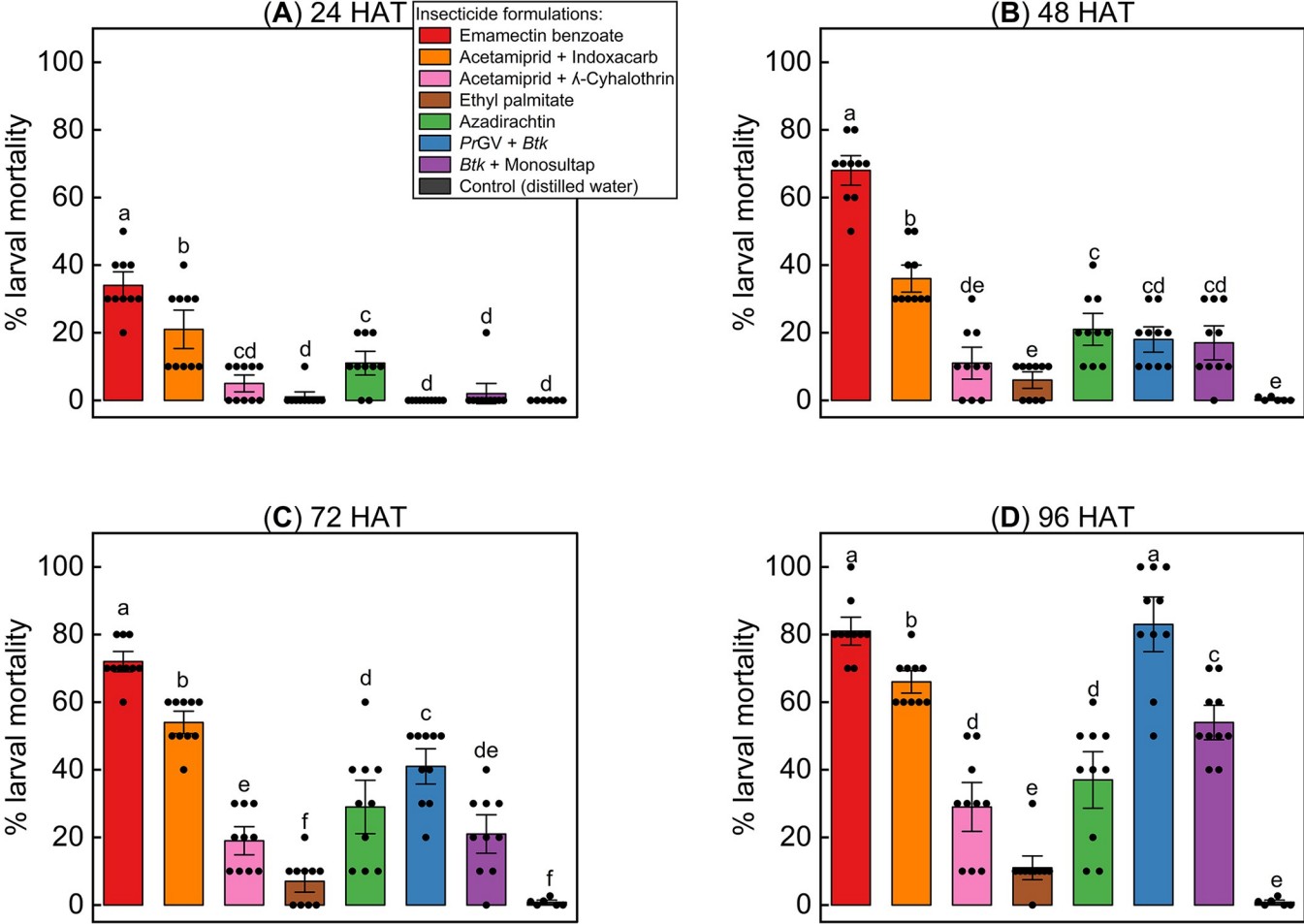

**Fig 1. Effects of manufacture recommended concentrations on the mortality rate of *Spodoptera frugiperda*.** Bars represent the means ± SE larval mortality at **(a)** 24 hours after treatment (HAT), **(b)** 48 HAT, **(c)** 72 HAT, and **(d)** 96 HAT. Different letters above bars indicate significant differences between treatments (Likelihood ratio test (LR test) applied to a generalized lineal model (GLM), followed by Tukey's honest significant difference (HSD) at $P < 0.05$).

Mortality rates were compared for different insecticide formulations using their recommended concentration at various time intervals (Fig 1). Emamectin benzoate had significantly higher mortality rates than other formulations at 24 HAT (LR test with GLM, $\chi^2 = 426.85$, $P < 0.0001$; Fig 1A). Emamectin benzoate and acetamiprid + indoxacarb had significantly higher mortality rates at 48 HAT (LR with GLM, $\chi^2 = 952.05$, $P < 0.0001$; Fig 1B). Emamectin benzoate, acetamiprid + indoxacarb, and *Pr*GV + *Btk* had the highest mortality rates at 72 HAT (LR with GLM, $\chi^2 = 860.78$, $P < 0.0001$; Fig 1C). *Pr*GV + *Btk* and emamectin benzoate had the highest mortality rates at 96 HAT (LR with GLM, $\chi^2 = 915.96$, $P < 0.0001$; Fig 1D). Emamectin benzoate and acetamiprid + indoxacarb had the highest mortality rates between 24–72 HAT, while *Pr*GV + *Btk* and *Btk* + monosultap had a progressive effect, with a significantly higher mortality rate at 96 HAT (LR with GLM, $\chi^2 = 915.96$, $P < 0.0001$; Fig 1).

### Sublethal effects on $F_0$ generation parent bionomics

Pupal duration, pupal mortality, adult longevity, and female fecundity of the F0 generation of *S. frugiperda* were all affected by the sublethal concentration of the tested insecticide formulations (LR with GLM, $P < 0.0001$; Table 3). For instance, pupal duration was significantly

**Table 3. Effect of sublethal concentrations on parent adults (F0 generation) of *Spodoptera frugiperda* (Mean ± SE).**

| Insecticide | Pupal duration (day) | % Mortality (Pupa–Adult) | Male longevity (day) | Female longevity (day) | Fecundity |
|---|---|---|---|---|---|
| Emamectin benzoate | 5.82±0.20[c] | 27.78±6.15[ab] | 8.40±0.47[d] | 8.95±0.51[c] | 238.00±13.65[bc] |
| Acetamiprid + Indoxacarb | 6.02±0.20[c] | 26.79±5.97[ab] | 9.86±0.46[bc] | 11.90±0.50[ab] | 150.50±09.27[cd] |
| Acetamiprid + ʎ-Cyhalothrin | 5.82±0.20[c] | 26.42±6.11[ab] | 9.19±0.46[cd] | 10.00±0.53[bc] | 248.37±32.76[b] |
| Ethyl palmitate | 9.45±0.19[a] | 34.33±5.84[b] | 9.67±0.46[bcd] | 8.87±0.47[c] | 187.12±12.88[bcd] |
| Azadirachtin | 6.37±0.23[c] | 30.00±6.55[ab] | 9.83±0.50[bc] | 10.47±0.54[bc] | 233.75±22.99[bc] |
| *Pr*GV + *Btk* | 9.36±0.19[a] | 27.87±5.79[ab] | 8.91±0.44[cd] | 8.67±0.49[c] | 115.75±05.32[d] |
| *Btk* + Monosultap | 7.21±0.21[b] | 32.26±5.99[b] | 10.94±0.50[b] | 10.54±0.46[bc] | 259.25±21.54[b] |
| Control | 7.44±0.18[b] | 8.06±3.49[a] | 12.85±0.41[a] | 13.26±0.41[a] | 580.87±34.38[a] |
| $\chi^2$ | 89.6 | 23.37 | 31.64 | 42.08 | 43.55 |
| *df* | 7 | 7 | 7 | 7 | 7 |
| *P* < **0.05** | < 0.0001 | 0.0014 | < 0.0001 | < 0.0001 | < 0.0001 |

Different letters within the same column represent significant differences at $P < 0.05$ (Likelihood Ratio test (LR) applied to a generalized linear model (GLM), followed by Tukey's HSD, test). Fecundity: Means number of eggs laid by individual female insect.

lengthened by *Pr*GV + *Btk* and ethyl palmitate, while shortened by azadirachtin, acetamiprid + ʎ-cyhalothrin, acetamiprid + indoxacarb, and emamectin benzoate, compared to the control treatment (LR with GLM, $P < 0.0001$; Table 3). The lowest fecundity was recorded with *Pr*GV + *Btk*, acetamiprid + indoxacarb, and ethyl palmitate treatments (LR with GLM, $P < 0.0001$; Table 3).

## Transgenerational sublethal effects on *S. frugiperda* bionomics

**Stage duration of F1 generation.** Compared to the control treatment, insecticides significantly reduced the stage length of F1 generation of *S. frugiperda* (paired bootstrap test, $P < 0.05$; Table 4; S1–S7 Tables). Azadirachtin (S2 Table) and ethyl palmitate (S3 Table) lengthened the egg stage, but *Pr*GV + *Btk* (S4 Table) shorten it compared to the control treatment (paired bootstrap test, $P < 0.05$; Table 4; S1–S4 Tables). Azadirachtin (S2 Table) and ethyl palmitate (S3 Table) lengthened the larval stage, but *Pr*GV + *Btk* (S4 Table), emamectin benzoate (S5 Table), and acetamiprid + indoxacarb (S6 Table) significantly shortened it (paired bootstrap test, $P < 0.05$; Table 4; S2–S6 Table). Emamectin benzoate (S5 Table) and acetamiprid + indoxacarb (S6 Table) had the shortest preadult stage, whereas azadirachtin

**Table 4. Response of duration (in day) of developmental stages of *Spodoptera frugiperda* to sublethal insecticide concentrations (Mean ± SE).**

| Insecticide | Egg | Larva | Pupa | Preadult | Male adult | Female adult |
|---|---|---|---|---|---|---|
| Emamectin benzoate | 2.83±0.07[bcd] | 10.35±0.22[d] | 5.76±0.11[f] | 19.31±0.25[e] | 9.90±0.26[cd] | 10.57±0.27[c] |
| Acetamiprid + Indoxacarb | 2.80±0.08[cd] | 10.35±0.21[d] | 5.94±0.11[f] | 19.11±0.30[e] | 12.09±0.34[b] | 12.97±6.16[a] |
| Acetamiprid + ʎ-Cyhalothrin | 2.88±0.08[bcd] | 11.10±0.26[c] | 6.32±0.13[e] | 20.45±0.32[d] | 10.62±0.31[c] | 11.00±0.35[c] |
| Ethyl palmitate | 3.28±0.10[a] | 15.31±0.40[a] | 7.45±1.27[b] | 26.41±0.45[a] | 7.00±0.15[e] | 7.15±0.18[e] |
| Azadirachtin | 3.04±0.08[a] | 15.08±0.37[a] | 6.80±0.19[c] | 25.10±0.42[b] | 7.23±0.32[e] | 7.45±0.33[e] |
| *Pr*GV + *Btk* | 2.67±0.08[d] | 10.41±0.15[d] | 8.77±0.17[a] | 21.97±0.24[c] | 9.97±0.35[c] | 10.33±0.37[d] |
| Btk + Monosultap | 2.74±0.09[d] | 11.19±0.23[c] | 6.62±0.14[d] | 20.57±0.32[d] | 12.48±0.49[a] | 12.68±0.43[ab] |
| Control | 3.00±0.07[b] | 13.96±0.10[b] | 7.22±0.12[b] | 24.24±0.16[b] | 13.31±0.27[a] | 13.49±0.35[a] |

The data in the table are mean (days) ± SE. Different superscript letters indicate significant difference $P < 0.05$ (paired bootstrap test with TWOSEX-MSChart 2023 software [35]). Consult S1–S8 Tables for an in-depth view of the data that informed these analyses.

**Table 5. Effect of sublethal insecticide concentrations on reproduction parameters of *Spodoptera frugiperda* (Mean ± SE).**

| Insecticide | Fecundity | TPOP (day) | Oviposition days | Mean Longevity |
|---|---|---|---|---|
| **Emamectin benzoate** | 237.70±09.19[a] | 19.53±0.41[e] | 5.16±0.14[d] | 25.25±0.94[g] |
| **Acetamiprid + Indoxacarb** | 276.97±10.23[a] | 19.75±0.44[e] | 5.27±0.17[d] | 27.11±1.04[de] |
| **Acetamiprid + ʎ-Cyhalothrin** | 295.19±10.65[a] | 21.46±0.43[cd] | 5.11±0.14[d] | 26.66±0.96[def] |
| **Ethyl palmitate** | 232.50±08.29[a] | 26.56±5.06[a] | 5.06±0.11[d] | 30.44±0.91[b] |
| **Azadirachtin** | 302.55±16.67[a] | 25.21±0.60[a] | 5.86±0.22[c] | 28.75±0.94[bc] |
| ***Pr*GV + *Btk*** | 350.43±17.95[a] | 22.37±0.32[c] | 6.67±0.29[b] | 26.37±1.14[cd] |
| ***Btk* + Monosultap** | 407.61±14.61[a] | 21.61±0.44[c] | 7.68±0.21[a] | 28.81±1.08[b] |
| **Control** | 428.82±12.69[a] | 24.69±0.22[b] | 7.05±0.13[b] | 33.36±1.21[a] |

TPOP: Total pre-oviposition period. The data in the table are mean values ± SE. Different superscript letters indicate significant difference $P < 0.05$ (paired bootstrap test with TWOSEX-MSChart 2023 software [35]). Consult S1–S8 Tables for an in-depth view of the data that informed these analyses.

(S2 Table) and ethyl palmitate (S3 Table) had the longest (paired bootstrap test, $P < 0.05$; Table 4). Azadirachtin, ethyl palmitate, *Pr*GV + *Btk*, and emamectin benzoate treatments had the shortest adult stages (paired bootstrap test, $P < 0.05$; Table 4; S2–S5 Tables).

**Reproduction parameters of F1 generation.** The insecticide treatments did not significantly affect the fecundity of F1 generation of *S. frugiperda* (paired bootstrap test, $P > 0.05$; Table 5; S1–S8 Tables). However, they had a significant impact on the total pre-oviposition period (TPOP), with and azadirachtin (S2 Table) and ethyl palmitate (S3 Table) having the longest TPOP, and emamectin benzoate (S5 Table) and acetamiprid + indoxacarb (S6 Table) having the shortest (paired bootstrap test, $P < 0.05$; Table 5; S2–S6 Tables). Additionally, the number of oviposition days was significantly reduced by ethyl palmitate (S3 Table), emamectin benzoate (S5 Table), acetamiprid + indoxacarb (S6 Table), acetamiprid + ʎ-cyhalothrin (S7 Table) (paired bootstrap test, $P < 0.05$; Table 5; S3–S7 Tables). *Pr*GV + *Btk* (S4 Table), emamectin benzoate (S5 Table), acetamiprid + indoxacarb (S6 Table), and acetamiprid + ʎ-cyhalothrin (S7 Table) had a significantly negative effect on the longevity of *S. frugiperda* compared to other treatments (paired bootstrap test, $P < 0.05$; Table 5; S4–S7 Tables).

**Population growth parameters of *Spodoptera frugiperda* at F1 generation.** The population growth parameters of F1 generation were significantly affected by the insecticide treatments, with azadirachtin (S2 Table) and ethyl palmitate (S3 Table) having negative effect on the Intrinsic Rate of Increase (r) of *S. frugiperda* (paired bootstrap test, $P < 0.05$; Table 6;

**Table 6. Effect of sublethal insecticide concentrations on the population parameters of F1 generation of *Spodoptera frugiperda* (Mean ± SE).**

| Insecticide | Intrinsic Rate of Increase (r in days) | Finite Rate of Increase (λ in days) | Net Reproductive Rate (R0) | Mean Generation Time (T in days) |
|---|---|---|---|---|
| **Emamectin benzoate** | 0.20±0.008[abc] | 1.22±0.010[abc] | 81.03±12.39[e] | 21.88±0.42[e] |
| **Acetamiprid + Indoxacarb** | 0.21±0.008[a] | 1.24±0.010[a] | 108.80±15.28[cde] | 21.73±0.49[e] |
| **Acetamiprid + ʎ-Cyhalothrin** | 0.20±0.007[ab] | 1.22±0.009[ab] | 117.44±15.57[bcd] | 23.40±0.45[d] |
| **Ethyl palmitate** | 0.16±0.006[g] | 1.18±0.008[g] | 101.34±13.54[cde] | 28.14±0.71[a] |
| **Azadirachtin** | 0.18±0.007[def] | 1.20±0.008[def] | 132.15±17.75[b] | 26.81±0.67[ab] |
| ***Pr*GV + *Btk*** | 0.19± 0.006[bcde] | 1.21±0.008[bcde] | 128.20±19.77[bc] | 25.15±0.34[c] |
| ***Btk* + Monosultap** | 0.20±0.006[a] | 1.23±0.009[a] | 164.10±23.58[a] | 24.62±0.38[c] |
| **Control** | 0.19±0.004[bcd] | 1.21±0.005[bcd] | 209.05±24.70[a] | 28.71±0.21[a] |

The data in the table are mean values ± SE. Different superscript letters indicate significant difference $P < 0.05$ (paired bootstrap test with TWOSEX-MSChart 2023 software [35]). Consult S1–S8 Tables for an in-depth view of the data that informed these analyses.

S2 and S3 Tables). These insecticides similarly affected the Finite Rate of Increase (Λ) of *S. frugiperda*, resulting in lowest values (paired bootstrap test, *P* < 0.05; Table 6; S2 and S3 Tables). The Net Reproductive Rate (Ro) of *S. frugiperda* was negatively significantly affected by ethyl palmitate (S3 Table), emamectin benzoate (S5 Table), acetamiprid + indoxacarb (S6 Table) (paired bootstrap test, *P* < 0.05; Table 6; S3–S6 Tables). The Mean Generation Time (T) was significantly reduced by *Pr*GV + *Btk* (S4 Table), emamectin benzoate (S5 Table), acetamiprid + indoxacarb (S6 Table), acetamiprid + ʎ-cyhalothrin (S7 Table), and *Btk* + monosultap (S8 Table) (paired bootstrap test, *P* < 0.05; Table 6; S4–S8 Tables).

**Age-stage survival of *Spodoptera frugiperda* at F1 generation.** Age-stage specific survival rate Sxj is the expected duration of neonate nymphs that will survive to age x and stage j. Fig 2 displays the effects of sublethal concentrations on Sxj of F1 generation of *S. frugiperda*, with no significant differences observed between the treatments (paired bootstrap test, *P* > 0.05; S1–S8 Tables). However, compared to the control treatment (0.85), the probability of neonate larvae reaching the adult stage was lower for all other treatments, with the lowest probability observed for *Pr*GV + *Btk* (0.73) (paired bootstrap test, *P* > 0.05; S1–S8 Tables).

## On-station experiments

### Minor rainy season

Throughout the minor rainy season experiment, the treatments had significant effect on the reduction of *S. frugiperda* larvae population per maize plant (One-way ANOVA, *N* = 10, *df* = 4, *P* < 0.05; Fig 3A–3D; S9 Table). The initial insecticidal spray administered 15 days after planting demonstrated higher population reduction with emamectin benzoate and *Pr*GV + *Btk* treatments compared to other treatment groups, observed 3 days after the insecticide treatment (DAT) (One-way ANOVA, *N* = 10, *df* = 4, *F* = 8.56, *P* < 0.001; Fig 3A; S9 Table). Compared to the control treatment, emamectin benzoate treatment plot had significantly higher population reduction of 70.04% (Dunnet's test, *P* < 0.001; Fig 3A; S9 Table). Similarly, 7 DAT, significant difference was observed between the treatment plots regarding the number of alive *S. frugiperda* larvae (One-way ANOVA, *N* = 10, *df* = 4, *F* = 19.82, *P* < 0.0001; Fig 3B; S9 Table). As a result, the population reductions on the emamectin benzoate (72.96%) and *Pr*GV + *Btk* (71.42%) treatment plots were significantly higher compared to the control plots (Dunnet's test, *P* < 0.001; Fig 3B; S9 Table).

After the second spray at 3 DAT, none of the insecticide treatments showed a significant difference compared to the control treatment based on Dunnett's test (*P* > 0.05; Fig 3C; S9 Table). However, at 7 DAT, significantly higher population reduction rates of 77.29% for the emamectin benzoate treatment and 66.62% for the *Pr*GV + *Btk* treatment were recorded, compared to the control treatment (Dunnett's test, *P* < 0.01; Fig 3D; S9 Table).

**Dry season.** During the dry season experiment, a significant difference was observed among the populations of alive larvae of *S. frugiperda* following insecticide sprays (Fig 4; S10 Table). After the first spray, at 3 DAT, the emamectin benzoate treatment plots demonstrated the highest reduction in the population of *S. frugiperda* larvae (One-way ANOVA, *N* = 10, *df* = 4, *F* = 14.64, *P* < 0.0001; Fig 4A; S10 Table), i.e., 58.33% higher than on control plots (Dunnett's test, *P* < 0.0001; Fig 4A; S10 Table). Furthermore, at 7 DAT, there were significantly higher population reductions on the emamectin benzoate (60.54%) and *Pr*GV + *Btk* (48.35%) treatment plots compared to the control plots., as determined by Dunnett's test (*P* < 0.01; Fig 4B; S10 Table).

Following the second spray, at 3 DAT, there was no significant difference observed between the treatments in terms of *S. frugiperda* larvae populations (One-way ANOVA, *N* = 10, *df* = 4, *F* = 0.59, *P* > 0.05; Fig 4C; S10 Table). However, at 7 DAT, the highest population reduction

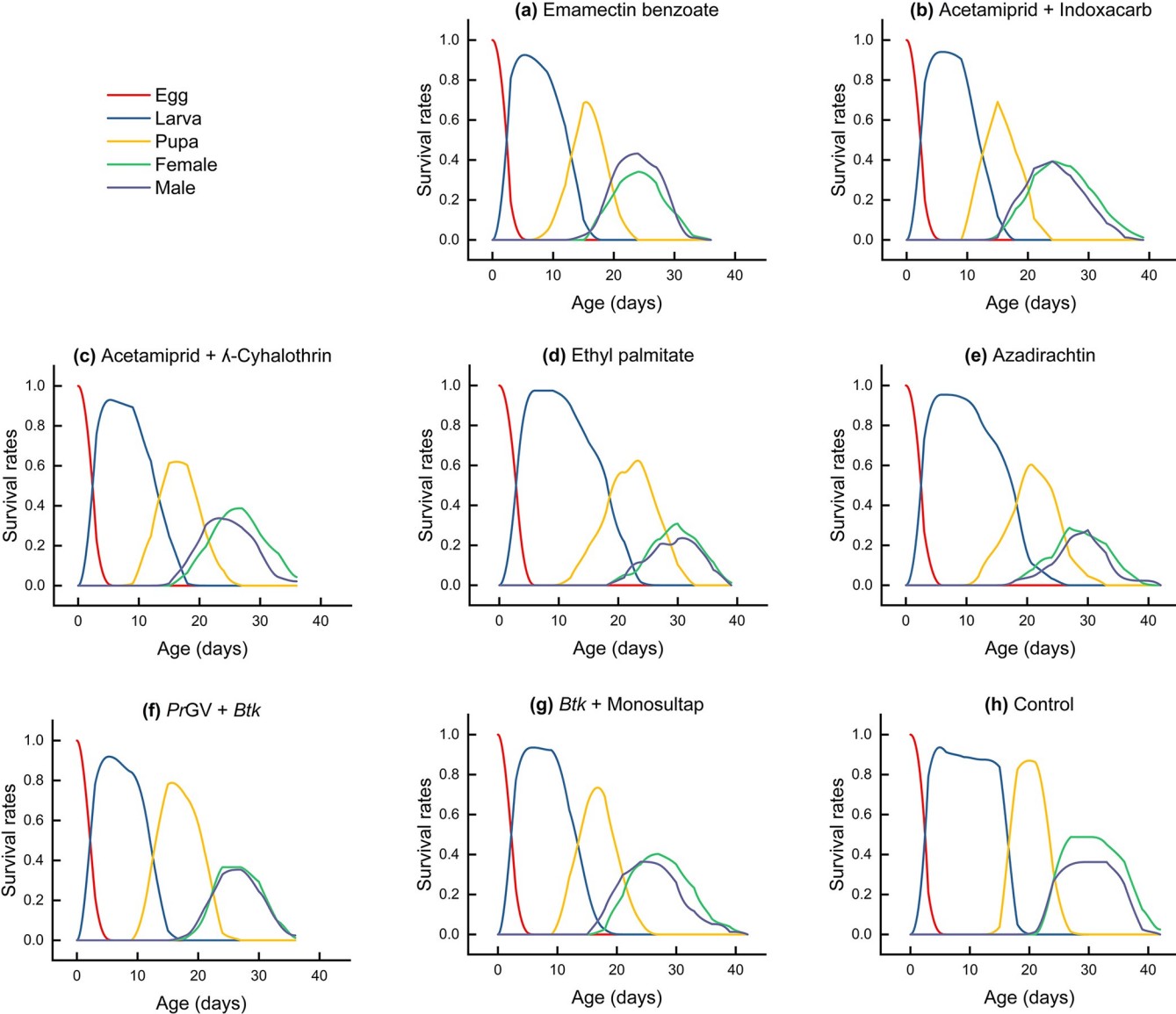

**Fig 2. Age-stage survival rates (Sxj) of *Spodoptera frugiperda* on maize leaves treated with sublethal concentration LC$_{25}$ and control (untreated).** Sxj is the probability that a newborn egg will survive to age x and stage j. The data evaluation was performed using the paired bootstrap test via the TWOSEX-MSChart 2023 software [35]. Consult *S1–S8* Tables for an in-depth view of the data that informed these analyses.

rates of 46.10% and 44.49% were recorded in the emamectin benzoate and *Pr*GV + *Btk* treatment plots (One-way ANOVA, $N = 10$, $df = 4$, $F = 5.84$, $P < 0.05$; Fig 4D; S10 Table), and significantly different from the control treatment (Dunnett's test, $P < 0.05$; Fig 4D; S10 Table).

On the third day following the third application of insecticide, significant decrease in the population of *S. frugiperda* larvae was observed in the plots treated with emamectin benzoate and *Pr*GV + *Btk*, in comparison to other treatments (One-way ANOVA, $N = 10$, $df = 4$, $F = 3.30$, $P < 0.05$; Fig 4E; S10 Table). Seventh day following the third insecticide spray, the most significant reduction (77.29%) in population of *S. frugiperda* larvae was recorded on *Pr*GV + *Btk* treated plots compared to the other treatments (One-way ANOVA, $N = 10$, $df = 4$, $F = 6.62$, $P < 0.01$; Fig 4F; S10 Table). In general, only the emamectin benzoate (60.71%) and

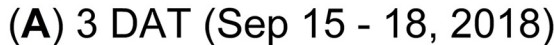

EB : Strike 1.9 EC™ (emamectin benzoate)
PR : Bypel 1 WP® (*Pr*GV + *Btk*)
BT : Agoo 55WP® (*Btk* + monosultap)
AZ : NeemGold 0.3 SC® (azadirachtin)
CT : Control (no insecticide)

- - - Population reduction on control plots

### First spray (Sep 15, 2018)

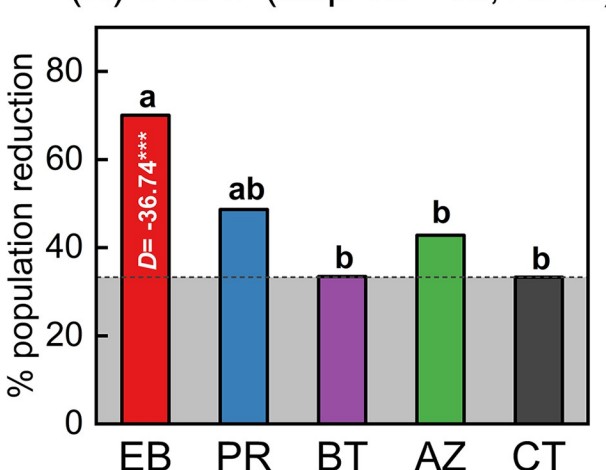

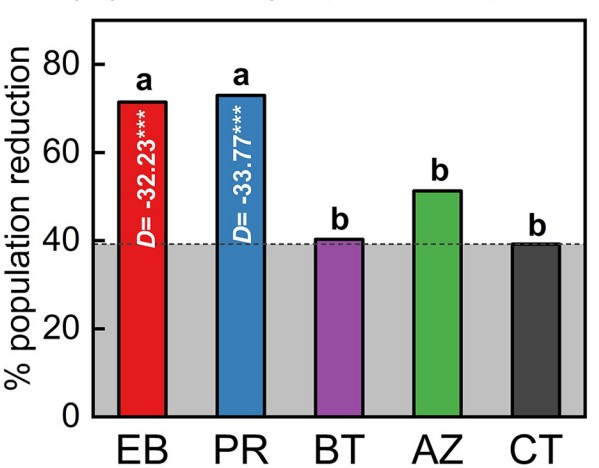

### Second spray (Oct 4, 2018)

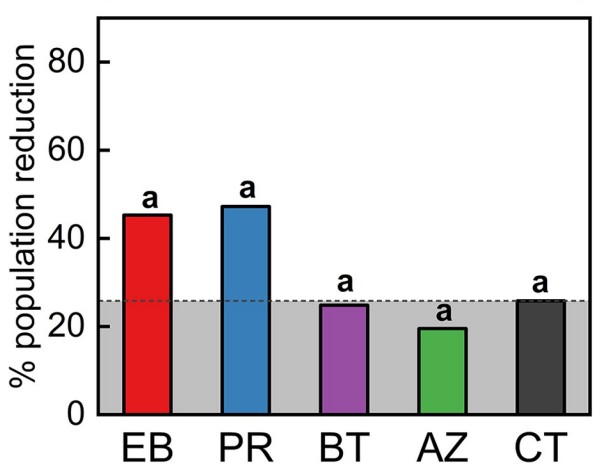

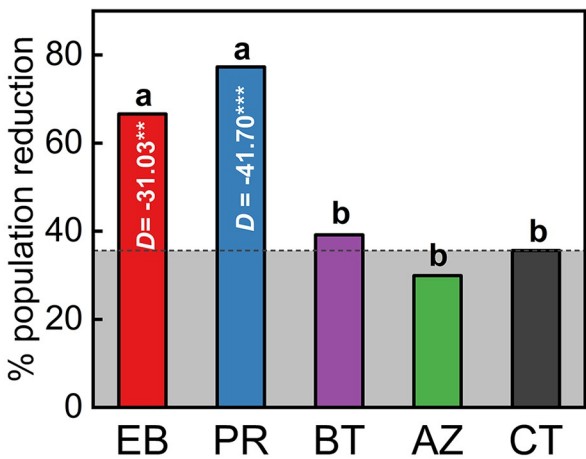

**Fig 3. Population reduction of *Spodoptera frugiperda* larvae during the minor rainy season on-station trial.** Two insecticide spray events were conducted: The first spray on September 15[th], 2018 (**A**, **B**), and the second spray on October 4[th], 2018 (**C**, **D**). The bars in the figure represent the mean population reduction, and the asterisks (*$P < 0.05$; **$P < 0.01$ and ***$P < 0.001$) above the bars indicate the significant difference between the insecticide-treated plots and the control using One-way ANOVA and Dunnett's Test ($\alpha = 0.05$). The term "pop." is an abbreviation for the population of live *S. frugiperda* larvae, while "DAT" indicates the count of days post-treatment. Refer to S9 Table for a detailed breakdown of the data.

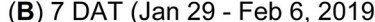

EB : Strike 1.9 EC™ (emamectin benzoate)
PR : Bypel 1 WP® (PrGV + Btk)
BT : Agoo 55WP® (Btk + monosultap)
AZ : NeemGold 0.3 SC® (azadirachtin)
CT : Control (no insecticide)
------- population reduction on control plots

**First insecticide spray (Jan 29, 2019)**

(**A**) 3 DAT (Jan 29 - Feb 1, 2019)

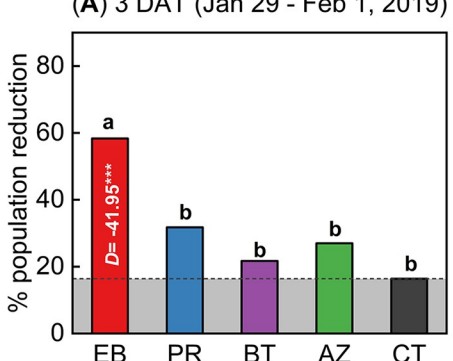

(**B**) 7 DAT (Jan 29 - Feb 6, 2019)

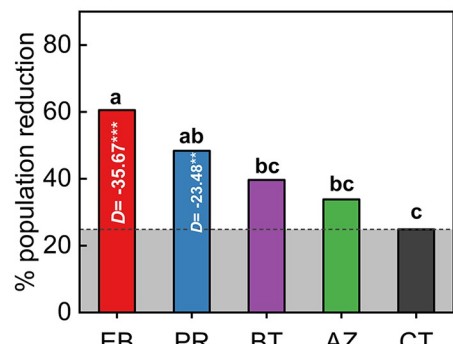

**Second insecticide spray (Feb 13, 2019)**

(**C**) 3 DAT (Feb 13 - 16, 2019)

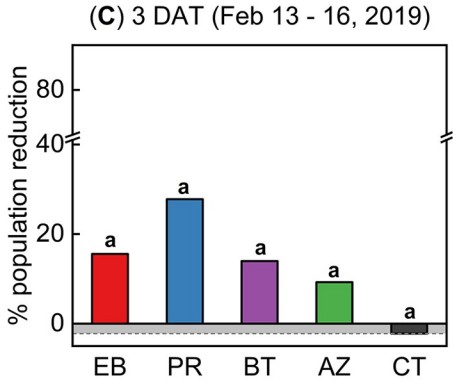

(**D**) 7 DAT (Feb 13 - 20, 2019)

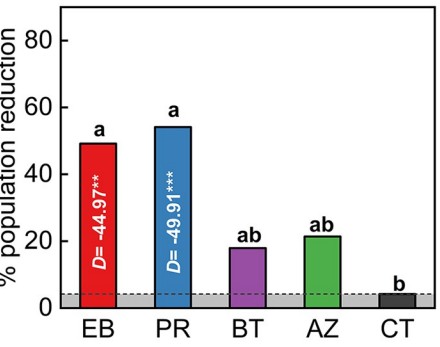

**Third insecticide spray (Feb 27, 2019)**

(**E**) 3 DAT (Feb 27 - Mar 2, 2019)

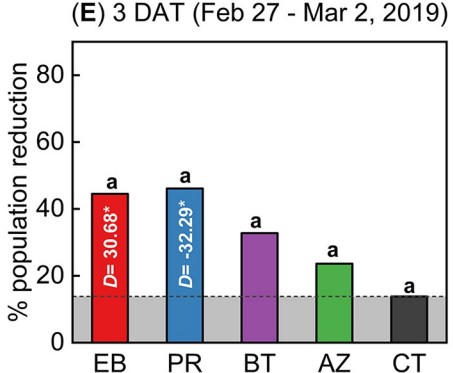

(**F**) 7 DAT (Feb 27 - Mar 6, 2019)

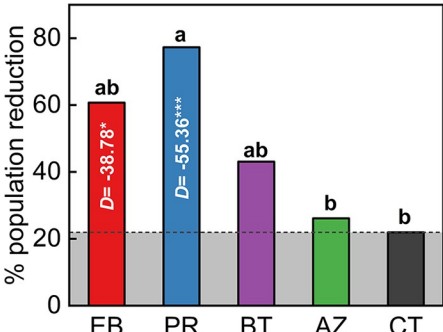

**Fig 4. Population reduction of *Spodoptera frugiperda* larvae during the dry season on-station trial.** Three insecticide spray events were conducted: The first spray on January 29[th], 2019 (**A**, **B**), the second spray on February 13[th], 2019 (**C**, **D**), and the third spray on February 27[th], 2019 (**E**, **F**). The bars in the figure represent the mean population reduction, and the asterisks (*$P < 0.05$; **$P < 0.01$ and ***$P < 0.001$) above the bars indicate the significant difference between the insecticide-treated plots and the control using One-way ANOVA and Dunnett's Test ($\alpha = 0.05$). The term "pop." is an abbreviation for the population of live *S. frugiperda* larvae, while "DAT" indicates the count of days post-treatment. Refer to S10 Table for a detailed breakdown of the data.

*Pr*GV + *Btk* treatments showed significantly higher population reduction compared to the control treatment at both 3 and 7 DAT (Dunnett's test, $P < 0.05$; Fig 4E and 4F; S10 Table).

**Dry grain yield of maize during on-station experiments.**   The study found that applying insecticide treatments had a positive impact on maize grain yield (Fig 5). During the minor rainy season, there was a significant difference in grain yields among the treatments. Maize plants treated with *Pr*GV + *Btk* and emamectin benzoate had the highest yields, while the lowest yields were recorded on *Btk* + monosultap plots (One-way ANOVA, $df = 4$, $F = 70.12$, $P < 0.0001$; Fig 5A; S11 Table). Similarly, during the dry season, *Pr*GV + *Btk* and emamectin benzoate significantly produced the highest maize grain yields (One-way ANOVA, $df = 4$, $F = 59.97$, $P < 0.0001$; Fig 5B; S11 Table). However, the azadirachtin treatment had the lowest yields among the insecticide-treated maize plants (One-way ANOVA, $df = 4$, $F = 59.97$, $P < 0.0001$; Fig 5B; S11 Table).

## Discussion

The study conducted toxicity bioassays on *Spodoptera frugiperda* larvae using various insecticide classes, including synthetic, botanical, and microbial insecticides. The results showed that emamectin benzoate had the highest larvicidal potency, whereas ethyl palmitate had the least. Previous research found substantial larval mortality in Lepidoptera larvae, especially those fed insecticide-treated diets [39–42]. Emamectin benzoate showed high toxicity to Lepidopterans such as the diamondback moth (*Plutella xylostella*, Plutellidae), the tomato leafminer (*Phthorimaea absoluta*, Gelechiidae), and *Spodoptera* sp. [31,41–44]. For instance, [31] and [41] found 0.0051 mg/L (expressed in ppm) and 0.0023–3.303 mg/L as $LC_{50}$ for emamectin benzoate on first and second early instar larvae of *S. frugiperda* in a leaf-dip bioassay, respectively, while [40] found 0.0014 mg/L for neonate larvae of *S. littoralis* at 48 h post-exposure. Though somewhat different, mean $LC_{50}$ for emamectin benzoate (0.019 mg/L) in our study is within the range established by [31]. These discrepancies in results may be attributed to variations in biological resources, species, age, methods, and exposure time. Nevertheless, our findings, along with other studies, support the notion that emamectin benzoate is a promising candidate for managing Lepidopteran pests, particularly defoliators like *S. frugiperda* [5,41]. Notably, the $LC_{50}$ values of emamectin benzoate, acetamiprid + indoxacarb, *Pr*GV + *Btk*, and *Btk* + monosultap in our study were lower than the manufacturer-recommended dosages, indicating that these insecticides could be effective in controlling *S. frugiperda* under field conditions using the concentrations suggested by the manufacturers.

Comparing the manufacture recommended concentrations of the insecticides, we found that effectiveness of the binary microbial *Pr*GV + *Btk* was slow in action compared to emamectin benzoate, and acetamiprid + indoxacarb but became more effective overtime, mainly at 96 HAT. Yet, little evidences exist from previous studies on the toxicity of *Pr*GV against *S. frugiperda*, while, its companion compound *Btk* toxin has been studied and used against a wide range of Lepidopteran species, including *S. frugiperda* [45–47]. Studies have, however been reported about the insecticidal activity of *Pr*GV on *Pieris rapae* Linnaeus (Lepidoptera: Pieridae) in vitro [48]. However, since natural entomopathogenic viruses utilized as biological control agents are species-specific, narrow-spectrum insecticides, their toxicity to other arthropod species (pests or beneficials) is essential. Meanwhile, natural entomopathogenic viruses have been shown to be an effective alternative to broad-spectrum insecticides [5,49,50].

Our study suggests that insecticidal efficacy testing should not only focus on determining lethal dosages but also include sublethal effects of formulations, especially with entomopathogens and botanicals. Our study found that *Pr*GV + *Btk*, Btk + monosultap, as well as the botanicals ethyl palmitate and azadirachtin, had significant sublethal effects on fecundity, reducing

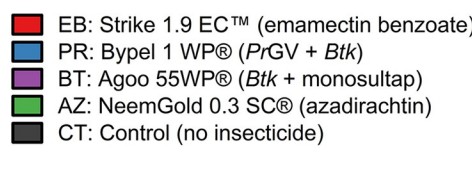

EB: Strike 1.9 EC™ (emamectin benzoate)
PR: Bypel 1 WP® (*Pr*GV + *Btk*)
BT: Agoo 55WP® (*Btk* + monosultap)
AZ: NeemGold 0.3 SC® (azadirachtin)
CT: Control (no insecticide)

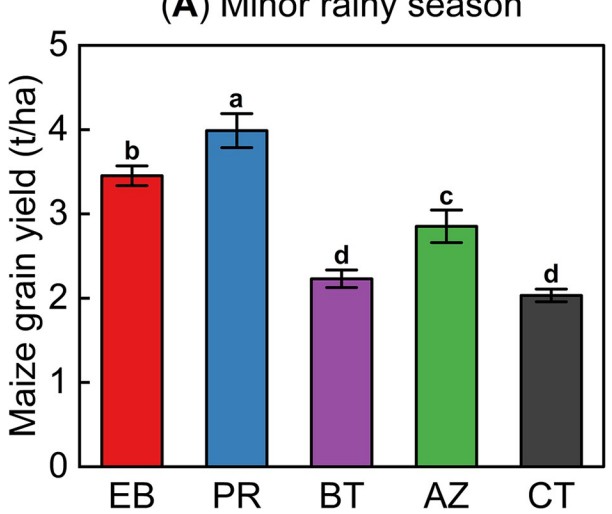

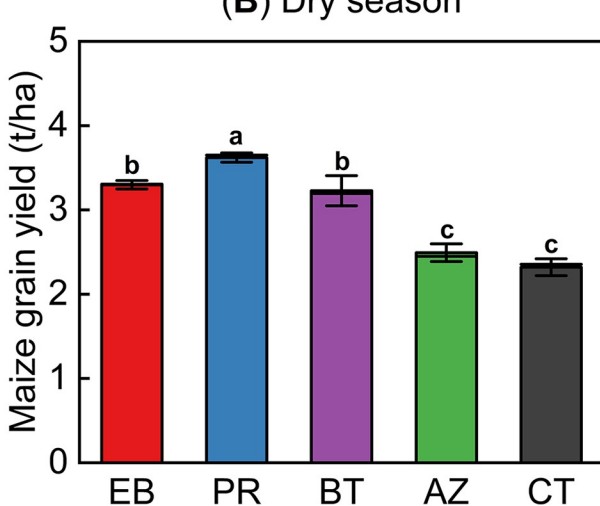

**Fig 5. Grain yield of maize during minor and dry season on-station experiments.** Bars represent the means ± SE of dry grain yields recorded on four plots (replicates). Different letters above bars indicate significant differences between treatments (One-way ANOVA followed by a Tukey's test, $P < 0.05$) during **(A)** the minor rainy and **(B)** dry season.

the net reproductive rate of both the *S. frugiperda* parents (F0 generation) and their offspring (F1 generation). Although these formulations showed weaker potency compared to synthetic insecticides during the toxicity bioassays, their sublethal effects are noteworthy. This finding is consistent with previous research [51,52]. For instance, it is known that azadirachtin commonly reduces fertility and offspring production in Lepidoptera adults [51]. In contrast, there is limited published research available on the insecticidal potency of ethyl palmitate as a stand-alone insecticide [53,54]. Ethyl palmitate has been used as a solvent or carrier for other insecticides and biopesticides, and there are a few studies investigating the efficacy of these formulations. Thus, our study constitutes a baseline for further investigations on the insecticidal role of ethyl palmitate and its mode of action.

The efficacy of *Btk* + monosultap and azadirachtin in controlling the incidence of *S. frugiperda* larvae showed inconsistency during the on-station experiments, unlike *Pr*GV + *Btk* and emamectin benzoate treatments. This suggests that the efficacy of *Btk* + monosultap and azadirachtin may be influenced by environmental factors such as temperature, UV radiation, and humidity [55,56]. However, it is worth noting that previous studies have reported the efficacy of *Btk* against *S. frugiperda* under field conditions [5,57,58]. It is important to consider that most of these studies utilized genetically engineered *Bt*-maize, which may explain the discrepancy between our findings and the previously reported efficacy of *Bt* in controlling *S. frugiperda*. Moreover, the field-evolved resistance of Lepidopteran pest species to *Bt*-based formulations [45,59]and the rapid development of resistance to monosultap in Lepidopteran species [60,61] can also explain the limited efficacy of *Btk* + monosultap against *S. frugiperda*. Furthermore, considering the $LC_{50}$ values, it is worth noting that they were higher than the recommended dosages applied for the *Btk* + monosultap and azadirachtin treatments in field conditions. Therefore, it is advisable to utilize higher dosages of these formulations to achieve

more effective control of *S. frugiperda*. On the other hand, the low efficacy in this study of aza-dirachtin-based formulations is due to the effects of manufacturing, storage, and transport conditions that can impact neem-based pesticides [62,63].

Consistently, *Pr*GV + *Btk* reduced the incidence of *S. frugiperda* and increased maize grain yield. While the *Pr*GV + *Btk*-based product has shown promising efficacy in both laboratory and on-station conditions, there is still a need for further investigations to fully understand its potential and serve as a relief for smallholder farmers. One important consideration is the need to confirm the complete profile of the commercial formulation used in our study. Conducting microbiological studies will be crucial in assessing the impact of the technical-grade *Pr*GV on target pests, understanding its mode of action, evaluating its persistence, and assessing any potential environmental effects [64]. These studies will provide valuable insights into the efficacy and safety of *Pr*GV + *Btk* and contribute to its appropriate use in pest management strategies.

Similar to *Pr*GV + *Btk*-based formulation, emamectin benzoate has demonstrated effectiveness against *S. frugiperda* in on-station conditions, and increased maize grain yield, consistent with findings from other studies [5,41,65]. However, it is important to acknowledge the high risk of resistance evolution in *S. frugiperda* populations exposed to emamectin benzoate [65]. Meanwhile, evidence of field-evolved resistance to emamectin benzoate has been reported in the native range of the pest [66]. Furthermore, it is crucial to investigate the potential impact of these formulations on non-target species, particularly natural enemies of *S. frugiperda* [5]. Understanding the compatibility of *Pr*GV + *Btk* and emamectin benzoate with beneficial organisms will help ensure the preservation of natural biological control agents, which are vital for sustainable pest management practices. To support the adoption of these formulations by local farmers, economic analyses are essential. Assessing the cost-effectiveness of *Pr*GV + *Btk* and emamectin benzoate-based formulations will provide valuable insights into their practicality and affordability for farmers, helping them make informed decisions regarding their use.

## Conclusion

Our study evaluated the toxicity of various insecticides against *Spodoptera frugiperda* larvae, and emamectin benzoate showed the highest larvicidal potency. The binary microbial *Pr*GV + *Btk* was effective but slow in action, while ethyl palmitate had the lowest potency. The study also found that insecticidal efficacy testing should not only focus on determining lethal dosages but also include sublethal effects of formulations. Although botanical and microbial formulations showed weaker potency compared to synthetic insecticides during the toxicity bioassays, their sublethal effects are noteworthy. However, the efficacy of azadirachtin and *Btk* + mono-sultap was inconsistent in on-station experiments, indicating that their effectiveness depends on environmental factors. Additionally, the study suggests that further investigations are needed to understand the mode of action of ethyl palmitate and *Pr*GV as a standalone insecticide. Overall, our study indicated that the semi-synthetic emamectin benzoate and the microbial *Pr*GV + *Btk* are good candidate in managing *S. frugiperda*.

## Supporting information

**S1 Table. Effects of the control treatment on the bionomics of *Spodoptera frugiperda*.**
(TXT)

**S2 Table. Impacts of sublethal doses of azadirachtin on the bionomics of *Spodoptera frugi-perda*.**
(TXT)

**S3 Table. Impacts of sublethal doses of ethyl palmitate on the bionomics of *Spodoptera frugiperda*.**
(TXT)

**S4 Table. Impacts of the combined sublethal doses of *Pieris rapae* Granulovirus and *Bacillus thuringiensis* subsp. *kurstaki* on the bionomics of *Spodoptera frugiperda*.**
(TXT)

**S5 Table. Impacts of sublethal doses of emamectin benzoate on the bionomics of *Spodoptera frugiperda*.**
(TXT)

**S6 Table. Impacts of the combined sublethal doses of acetamiprid and indoxacarb on the bionomics of *Spodoptera frugiperda*.**
(TXT)

**S7 Table. Impacts of the combined sublethal doses of acetamiprid and lambda-cyhalothrin on the bionomics of *Spodoptera frugiperda*.**
(TXT)

**S8 Table. Impacts of the combined sublethal doses of *Bacillus thuringiensis* subsp. *kurstaki* and Monosultap on the bionomics of *Spodoptera frugiperda*.**
(TXT)

**S9 Table. Number of alive *Spodoptera frugiperda* larvae per ten maize plants per plot during the minor rainy season on-station trial.** CT: Control (no insecticide); AZ: NeemGold 0.3 SCⓇ (azadirachtin); BT: Agoo 55WPⓇ (*Btk* + Monosultap); PR: Bypel 1 WPⓇ (*Pr*GV + *Btk*); EB: Strike 1.9 EC™ (emamectin benzoate).
(DOCX)

**S10 Table. Number of alive *Spodoptera frugiperda* larvae per ten maize plant per plot during the dry season on-station trial.** CT: Control (no insecticide); AZ: NeemGold 0.3 SCⓇ (azadirachtin); BT: Agoo 55WPⓇ (*Btk* + Monosultap); PR: Bypel 1 WPⓇ (*Pr*GV + *Btk*); EB: Strike 1.9 EC™ (emamectin benzoate).
(DOCX)

**S11 Table. Maize grain yield (t) per hectare (ha) during the minor rainy and dry seasons.** CT: Control (no insecticide); AZ: NeemGold 0.3 SCⓇ (azadirachtin); BT: Agoo 55WPⓇ (*Btk* + Monosultap); PR: Bypel 1 WPⓇ (*Pr*GV + *Btk*); EB: Strike 1.9 EC™ (emamectin benzoate).
(DOCX)

## Acknowledgments

KRF warmly thanks the German Academic Exchange Service (DAAD) for the two-year MPhil Scholarship. We are grateful to the research technicians at the University of Ghana's SIREC, Mr. Tegbe R and Mr. Awittor RK for their supports during the on-station trials.

## Author Contributions

**Conceptualization:** Kokou Rodrigue Fiaboe, Ken Okwae Fening, Winfred Seth Kofi Gbewonyo.

**Data curation:** Kokou Rodrigue Fiaboe, Sharanabasappa Deshmukh.

**Formal analysis:** Kokou Rodrigue Fiaboe, Sharanabasappa Deshmukh.

**Project administration:** Ken Okwae Fening.

**Supervision:** Ken Okwae Fening, Winfred Seth Kofi Gbewonyo.

**Validation:** Ken Okwae Fening.

**Writing – original draft:** Kokou Rodrigue Fiaboe.

**Writing – review & editing:** Kokou Rodrigue Fiaboe, Ken Okwae Fening, Winfred Seth Kofi Gbewonyo, Sharanabasappa Deshmukh.

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
