## [Decision Letter · Decision Letter 0]

10 May 2023

PONE-D-23-07348Bionomic responses of Spodoptera frugiperda (J. E. Smith) to lethal and sublethal concentrations of selected insecticidesPLOS ONE

Dear Dr. Fiaboe,

Thank you for submitting your manuscript to PLOS ONE. After careful consideration, we feel that it has merit but does not fully meet PLOS ONE’s publication criteria as it currently stands. Therefore, we invite you to submit a revised version of the manuscript that addresses the points raised during the review process.

Please improve the text of the manuscript according to the reviewers' comments.

We look forward to receiving your revised manuscript.

Kind regards,

Yonggen Lou

Academic Editor

PLOS ONE

“NO”

Reviewers' comments:

Reviewer's Responses to Questions

**Comments to the Author**

1. Is the manuscript technically sound, and do the data support the conclusions?

Reviewer #1: Partly

Reviewer #2: Yes

2. Has the statistical analysis been performed appropriately and rigorously? 

Reviewer #1: I Don't Know

Reviewer #2: Yes

3. Have the authors made all data underlying the findings in their manuscript fully available?

Reviewer #1: Yes

Reviewer #2: Yes

4. Is the manuscript presented in an intelligible fashion and written in standard English?

Reviewer #1: Yes

Reviewer #2: No

5. Review Comments to the Author

Reviewer #1: Review of the manuscript: Bionomic responses of Spodoptera frugiperda (J. E. Smith) to lethal and sublethal concentrations of selected insecticides

My overall comments are as follow:

1. The results are overall clearly presented.

2. Some citations should be added into the introduction.

3. Presentation of data from field experiments maybe not properly.

Brief information of the pesticides used in this study should be mentioned in the introduction.

Line 55: Please briefly introduce S. frugiperda. For example, polyphagous, strong preference to maize, resistance to pesticides, etc.

Line 115: Why castor bean leaves instead of maize leaves were used in the rearing? Different diets of S. frugiperda might alter the physiological status of this pest and therefore change their resistance to pesticides.

Line 225: Homogeneity of variance test and normality test should be mentioned before ANOVA analysis.

Line 233: full name of HAT (hours after treatment) should be mentioned in the first time

Table 2: the annotation of “N”

Table 3: “Fecundity” means number of eggs produced by individual female insect?

Table 7 and 8: Compared with number of alive larvae, it might be more reasonable to compare the survival rate of S. frugiperda larvae after pesticide spray?

Reviewer #2: Spodoptera frugiperda is one of the most important invasive insect and farmers often resort to the use hazardous pesticides. In this study, the effects of botanicals, microbial, and semi-synthetic insecticides in Ghana for controlling S. frugiperda were investigated. The authors found some interesting results that the emamectin benzoate has the highest acute larvicidal effect with lower LC50 values, sublethal effects of Pr GV + Btk, azadirachtin, and ethyl palmitate on the bionomics of S. frugiperda. The experiments in field showed semi-synthetic emamectin benzoate and the bioinsecticide PrGV + Btk are good candidates for managing S. frugiperda. The manuscript need to be minor revised before published on this journal. Some suggestion are as follows,

# The language of this paper need to be improved, it is better to reduce the qualitative statement in the result description through this study.

# The unit ppm could not indicate the connection of solution intuitively, please use mg/L or g/L.

# Study location is not necessary or this part could be introduced in methods in field.

# Line 170, Ten (10) or Line Two (2) revised to Ten or Two.

# Line 116, the relative humidity miss the percent sign % after the number.

6. PLOS authors have the option to publish the peer review history of their article (what does this mean?). If published, this will include your full peer review and any attached files.

Reviewer #1: No

Reviewer #2: No

---

## [Author Response · Author response to Decision Letter 0]

30 Jun 2023

Response to Reviewers

Reviewer #1

Comment 1: Brief information of the pesticides used in this study should be mentioned in the introduction.

Response 1: The introduction is modified with information on the pesticides used in the study according to the reviewer comment.

Comment 2: Line 55: Please briefly introduce S. frugiperda. For example, polyphagous, strong preference to maize, resistance to pesticides, etc.

Responses 2: Lines 56 – 63: Revision made according to the reviewer comment 

Comment 3: Line 115: Why castor bean leaves instead of maize leaves were used in the rearing? Different diets of S. frugiperda might alter the physiological status of this pest and therefore change their resistance to pesticides.

Response 3: In our research, the decision to rear S. frugiperda larvae on castor bean leaves instead of maize leaves was based on practical considerations. Castor bean leaves have been widely used as a suitable diet for rearing fall armyworm in laboratory settings due to their availability, nutritional value, and support for larval development. This choice allowed us to maintain a controlled and consistent rearing environment, ensuring uniformity in the developmental stage and physiological status of the tested insects. However, we acknowledge that there is a potential discrepancy between the rearing diet and the choice of maize leaves during the leaf-dip bioassays. It is possible that the use of different diets could influence the physiological responses and resistance mechanisms of S. frugiperda to the tested pesticides. However, the use of maize leaves consistently during the bioassays isolate any potential influences of the diet on the bioactivity of the insecticides, providing a clearer understanding of the insect's susceptibility to the tested pesticides.

Comment 4: Line 225: Homogeneity of variance test and normality test should be mentioned before ANOVA analysis.

Response 4: Lines 229 – 231: Statement on homogeneity of variance and normality tests is included according to the reviewer’s comment.

Comment 5: Line 233: full name of HAT (hours after treatment) should be mentioned in the first time.

Response 5: Line 243: “(hours after treatment)” mentioned after HAT 

Comment 6: Table 2: the annotation of “N”

Response 6: Line 264: “N” defined.

Comment 7: Table 3: “Fecundity” means number of eggs produced by individual female insect?

Response 7: Line 178: For clarity, the term “Fecundity” is defined when mentioned for the first time in the text.

Comment 8: Table 7 and 8: Compared with number of alive larvae, it might be more reasonable to compare the survival rate of S. frugiperda larvae after pesticide spray?

Response 7: Tables 7 and 8 are replaced with Figures 3 and 4, presenting the population reduction.

Reviewer #2: 

# The language of this paper needs to be improved; it is better to reduce the qualitative statement in the result description through this study.

Response: The language was revised accordingly.

# The unit ppm could not indicate the connection of solution intuitively, please use mg/L or g/L.

Response: ppm is replaced with mg/L according to review suggestion.

# Study location is not necessary or this part could be introduced in methods in field.

Response: “Study location” is relocated to on-station experiment section

# Line 170, Ten (10) or Line Two (2) revised to Ten or Two.

Response: Correction made accordingly.

# Line 116, the relative humidity misses the percent sign % after the number.

Response: Correction made accordingly.

---

## [Decision Letter · Decision Letter 1]

8 Aug 2023

Bionomic responses of Spodoptera frugiperda (J. E. Smith) to lethal and sublethal concentrations of selected insecticides

PONE-D-23-07348R1

Dear Dr. Fiaboe,

We’re pleased to inform you that your manuscript has been judged scientifically suitable for publication and will be formally accepted for publication once it meets all outstanding technical requirements.

Kind regards,

Yonggen Lou

Academic Editor

PLOS ONE

Additional Editor Comments (optional):

Reviewers' comments:

Reviewer's Responses to Questions

**Comments to the Author**

1. If the authors have adequately addressed your comments raised in a previous round of review and you feel that this manuscript is now acceptable for publication, you may indicate that here to bypass the “Comments to the Author” section, enter your conflict of interest statement in the “Confidential to Editor” section, and submit your "Accept" recommendation.

Reviewer #1: (No Response)

Reviewer #2: All comments have been addressed

2. Is the manuscript technically sound, and do the data support the conclusions?

Reviewer #1: Yes

Reviewer #2: Yes

3. Has the statistical analysis been performed appropriately and rigorously? 

Reviewer #1: Yes

Reviewer #2: Yes

4. Have the authors made all data underlying the findings in their manuscript fully available?

Reviewer #1: Yes

Reviewer #2: Yes

5. Is the manuscript presented in an intelligible fashion and written in standard English?

Reviewer #1: Yes

Reviewer #2: Yes

6. Review Comments to the Author

Reviewer #1: (No Response)

Reviewer #2: The manuscript is improved welly and more consistent with standard of this publication. Some minor suggestion,

1. please place two methods of "plant meterial" and "Insecticide preparation, application and data collection" under the method of "On-station experiments". or corresponding to "Laboratory studies", set a new titleline to summarize last three methods.

2. figure 1, the legend symbol frame need to be adjust

3. figure 3, 4, 5, the the legend symbol frame could be delleted.

7. PLOS authors have the option to publish the peer review history of their article (what does this mean?). If published, this will include your full peer review and any attached files.

Reviewer #1: No

Reviewer #2: No

---

## [Editor Report · Acceptance letter]

11 Aug 2023

PONE-D-23-07348R1 

Bionomic responses of *Spodoptera frugiperda *(J. E. Smith) to lethal and sublethal concentrations of selected insecticides 

Dear Dr. Fiaboe:

I'm pleased to inform you that your manuscript has been deemed suitable for publication in PLOS ONE. Congratulations! Your manuscript is now with our production department. 

Kind regards, 

on behalf of

Dr. Yonggen Lou 

Academic Editor

PLOS ONE